# A “Galactic” Chest X-ray

**DOI:** 10.3390/diagnostics11050899

**Published:** 2021-05-18

**Authors:** Cristiano Carbonelli, Angela de Matthaeis, Antonio Mirijello, Concetta Di Micco, Evaristo Maiello, Salvatore De Cosmo, Paolo Graziano

**Affiliations:** 1Unit of Pneumology, Department of Medical Sciences, IRCCS Casa Sollievo della Sofferenza, 71013 San Giovanni Rotondo, Italy; 2Unit of Internal Medicine, Department of Medical Sciences, IRCCS Casa Sollievo della Sofferenza, 71013 San Giovanni Rotondo, Italy; angeladematthaeis@gmail.com (A.d.M.); s.decosmo@operapadrepio.it (S.D.C.); 3Unit of Oncology, Department of Medical Sciences, IRCCS Casa Sollievo della Sofferenza, 71013 San Giovanni Rotondo, Italy; c.dimicco@operapadrepio.it (C.D.M.); e.maiello@operapadrepio.it (E.M.); 4Unit of Pathology, Department of Services, IRCCS Casa Sollievo della Sofferenza, 71013 San Giovanni Rotondo, Italy; p.graziano@operapadrepio.it

**Keywords:** miliary pulmonary involvement, interventional pulmonology, lung cancer, metastasis

## Abstract

Clinical manifestations accompanying respiratory failure with insidious and rapidly progressive onset are often non-specific. Symptoms such as a cough, dyspnea, and fever are common to a large number of inflammatory, infectious, or neoplastic diseases. During the COVID-19 pandemic it is essential to limit the use of hospital services and inappropriate diagnostic techniques. A particular radiological pattern can orient the clinical and laboratory scenario and guide the diagnostic workup. A 58-year-old woman was admitted to our COVID-19 unit for suspected coronavirus infection. She was complaining of worsening dyspnea, tachycardia, and low grade fever. A chest X-ray showed diffuse, alveolar, and interstitial lung involvement with micronodules tending to coalescence. This radiographic pattern known as “galaxy sign”, consistent with diffuse, coalescing nodular miliary pulmonary involvement, simulating a non-specific alveolar opacification of the lungs is typical of a few pneumological differential diagnoses, represented by sarcoidosis, tuberculosis, pneumoconiosis, and metastatic lesions, and virtually excludes an interstitial viral pneumonitis. The use of endoscopic techniques can, in such cases, confirm the clinical suspicion for initiating appropriate targeted therapies.

A 58-year-old woman complaining of dyspnea at rest, tachycardia and oxyhemoglobinemic desaturation was transferred from a peripheral hospital to our COVID-19 unit for suspected coronavirus infection. Room air oxygen saturation (SpO_2_) was 84%, heart rate was 111 bpm, blood pressure was 130/85 mmHg, body temperature was 36.6 °C. No lung sounds were detected at thoracic auscultation. Past medical history was unremarkable: the patient had never smoked nor used drugs and she was a housewife. A few days earlier, dyspnea increased along with low-grade fever, and the patient was rapidly worsening. EKG showed sinus tachycardia. The administration of a fraction of inspired oxygen of 50%, produced an improvement of SpO_2_ to 94%, but severe exertional dyspnea was persisting. Despite a negative nasopharyngeal swab for COVID-19, clinical and radiological findings raised a high suspicion of coronavirus infection and the need for isolating the patient into a COVID-19 unit.

Table 1 illustrates the biochemical workup and blood gas analysis. Interestingly, despite the elevation of erythrocyte sedimentation rate, white blood cells, platelets and fibrinogen, C reactive protein was normal (Table 1).

A chest radiograph showed diffuse, alveolar and interstitial lung involvement with micronodules tending to coalescence (Figure 1). This radiographic pattern, known as “galaxy sign”, consistent with diffuse, coalescing nodular miliary pulmonary involvement, simulating a mass-like region or a non-specific alveolar opacification of the lungs [1].
**What Would Be Your Next Diagnostic Examination?**

A bronchoscopy with bronchoalveolar lavage (BAL) and transbronchial pulmonary sampling was performed. SARS-COV-2 real-time reverse-transcriptase-polymerase-chain-reaction from BAL fluid was negative. In the absence of laboratory signs suggestive for infectious triggers, physicians should be aware that the “galaxy sign” is common to several diseases [2], the most common being sarcoidosis, tuberculosis, pneumoconiosis, and metastatic lesions [3].

Histopathological diagnosis of lung adenocarcinoma showing lepidic, non-mucinous, and acinar patterns (Figure 2), explained the nodular radiological pattern, while the subsequent demonstration of the c.2573T > G p. (Leu858Arg) mutation in exon 21 of the EGFR gene explained the whole remaining clinical scenario: a diffuse miliary oncological disease, often characterized by rapid progression, in a non-smoker patient [4].

Once microbiological etiology excluded, treatment with osimertinib, an oral EGFR tyrosine kinase inhibitor, was started. The clinical and radiological picture significantly improved, and after six months the patient is still continuing oncological follow-up.
**Conclusions**

The present case is paradigmatic for at least three reasons: (1) coalescent pulmonary micronodular involvement, simulating diffuse alveolar opacification, should be further evaluated for a differential diagnosis also including treatable oncological conditions; (2) in the current context of a COVID-19 pandemic, endoscopic procedures might be crucial to rule out infections and to prevent the access of potentially contagious patients to non-infective wards [5]; (3) compared to the poor sensitivity of the BAL alone, endoscopic techniques should include transbronchial biopsies to assure best adequacy of molecular analysis [6].

## Figures and Tables

**Figure 1 diagnostics-11-00899-f001:**
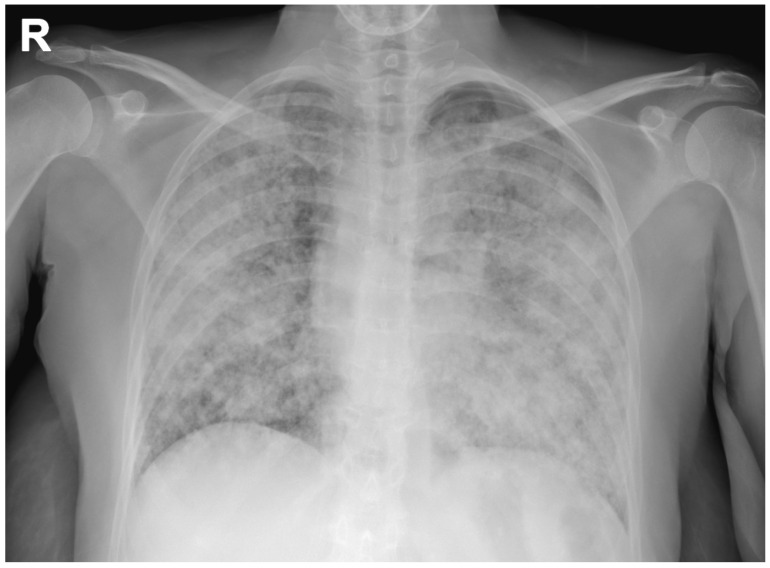
Chest radiograph showing a diffuse, alveolar and interstitial lung involvement with coalescing micronodules, simulating a non-specific alveolar opacification of the lungs (galaxy sign) (R: right side).

**Figure 2 diagnostics-11-00899-f002:**
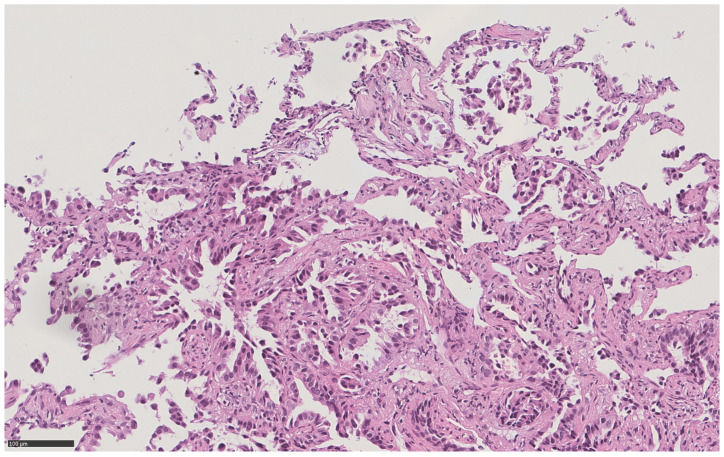
High power (21x) hematoxylin and eosin view of a bronchoscopic transbronchial pulmonary sample in the same patient, showing primary lung adenocarcinoma with lepidic non mucinous and acinar patterns.

**Table 1 diagnostics-11-00899-t001:** Results of laboratory tests and blood gas analysis.

Laboratory Tests	Results	Reference Range
ALT (U/L)	20.00	10–42
AST (U/L)	24.00	8–30
Total bilirubin (mg/dL)	0.40	0.2–1.0
Sodium (mmol/L)	159.0	136–145
Potassium (mmol/L)	4.9	3.5–5.0
Clorum (mmol/L)	125.0	98–107
Creatinine (mg/dL)	0.50	0.55–1.02
Urea (mg/dL)	30.0	15–38
Glycemia (mg/dL)	92.0	70–100
Total protein (g/dL)	6.30	6.4–8.2
C-reactive protein (mg/dL)	<0.290	<0.30
Erythrocyte sedimentation rate (mm)	45	2–15
Procalcitonin (µ/L)	0.09	<0.50
Hemoglobin (g/dL)	14.1	12–16
Mean corpuscular volume (FL)	87	77–98
Red blood cells (10^3^/mm^3^)	5080	4200–5400
White blood cells (/mm^3^)	20,650	4300–10,800
Neutrophilis (%)	94.6	40–80
Lymphocytes (%)	3.5	10–45
Monocytes (%)	1.3	2.0–10.0
Basophils (%)	0.00	<1.50
Eosinophils (%)	0.4	0.3–7.0
Platelets (10^3^/mm^3^)	481	130–400
Fibrinogen (mg/dL)	669	150–400
INR (Ratio)	1.03	0.80–1.20
pH	7.44	7.35–7.45
PCO_2_ (mmHg)	31.5	35.0–45.0
PO_2_ (mmHg)	52.9	80.0–100.0
HCO_3_ (mmol/L)	21.9	22.0–26.0
Lactate (mmol/L)	0.9	0.5–1.6
SpO_2_ (%)	91.2	90–99

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
