# Peer review of "A “Galactic” Chest X-ray"

_diagnostics, 2021, doi:10.3390/diagnostics11050899_

Round 1

Reviewer 1 Report

Dear Authors, I have received your case report on an unusual presentation of miliary metastatic lung carcinoma. I find this case to be of interest, and I would only suggest a few edits to the abstract/text.

Abstract

1) Please do include a summary of your case in this section instead of giving a general overview of the matter. You can easily very much shorten/summarize the initial part (lines 15 to 19) to gain space.

2) Line 19: do make explicit that you refer to the COVID-19 pandemic instead of writing "in the current pandemic era".

Text

3) Line 30 and following: you say that this woman was transferred to your "COVID-19 unit" (incidentally, please write COVID-19 all caps in all instances, as in line 36 and 70). But then you declare (line 36) that her swab was negative. Does it mean that she was admitted to a COVID-19 ward only because of suspected infection?

4) Line 70/71: the meaning of your "reason 2" is quite obscure and should be clarified/reworded. Indeed, if I understood correctly, in this case the endoscopic procedure "functioned" the other way round, i.e., preventing harmful/useless access (or at least, prolonged stay) of a patient with cancer to an infectious disease ward.

Author Response

Dear Editor,

we thank the Reviewer for his comments to our manuscript.

All the comments have been addressed in the text and we have attached a point-to-point-reply.

Sincerely,

Cristiano Carbonelli

Reviewer 2 Report

In my opinion, this appears to be a fairly well written paper, but some clarifications were required to improve the final quality of the article.

abstract 
page 1 line 18: the differential diagnosis cannot be based on "Physician's clinical confidence with the diseases", but rather on the clinical, laboratory and instrumental picture ... please reformulate the sentence.
page 1 line 24: the list of pathologies that can occur with the galaxy sign is quite vast, while here are only a few ... Please, at least insert sarcoidosis, of which the galaxy sign is a not uncommon sign of active disease.

keywords: they seem few, and not very specific: please expand the number of keywords and indicate more specific for the topic

Maintext

page 1 line 31: Was it a repeated nasopharyngeal swab during hospitalization? if so, was it positive or negative? this was not written, but it is important to know.

Page 3 line 55: again, the list of possible disease presented with galaxy sign is too small: please improve. as you have written a paper on the galaxy sign, it would be very useful to provide a list (and perhaps a table), ordered by frequency, complete and exhaustive of all the pathologies that can present with this sign on chest imaging.
Page 4 line 66: Has the imaging changed in any way?

Author Response

(The authors gave the same response as above.)

Reviewer 3 Report

As the authors point out in this manuscript, symptoms such as cough, dyspnea, and fever are common symptoms of many inflammatory, infectious, or neoplastic illnesses. The authors state that the chest radiographs shown in FIG. 1, the so-called galaxy signs, are typical of alveolar differential diagnosis represented by tuberculosis, pneumoconiosis, and metastatic lesions. Therefore, they also argue that radiography patterns alone should not determine the diagnostic approach, but that endoscopic techniques should be used (Fig. 2 shows bronchoscopy results).

I think their claim is correct and these images are very interesting. I think it is appropriate to publish this manuscript because I think it will be useful for urgently deciding the treatment policy even in the current COVID-19 pandemic era. However, if Table 1 is included, please explain the items further in the text.

Author Response

(The authors gave the same response as above.)
